# Kink far below the Fermi level reveals new electron-magnon scattering channel in Fe

E. Młyńczak [1,2], M.C.T.D. Müller[1,3], P. Gospodarič [1], T. Heider [1], I. Aguilera[1,3], G. Bihlmayer [1,3], M. Gehlmann[1], M. Jugovac [1], G. Zamborlini[1], C. Tusche [1], S. Suga[1,4], V. Feyer [1], L. Plucinski[1], C. Friedrich[1,3], S. Blügel [1,3] & C.M. Schneider [1]

Many properties of real materials can be modeled using ab initio methods within a single-particle picture. However, for an accurate theoretical treatment of excited states, it is necessary to describe electron-electron correlations including interactions with bosons: phonons, plasmons, or magnons. In this work, by comparing spin- and momentum-resolved photoemission spectroscopy measurements to many-body calculations carried out with a newly developed first-principles method, we show that a kink in the electronic band dispersion of a ferromagnetic material can occur at much deeper binding energies than expected ($E_b = 1.5$ eV). We demonstrate that the observed spectral signature reflects the formation of a many-body state that includes a photohole bound to a coherent superposition of renormalized spin-flip excitations. The existence of such a many-body state sheds new light on the physics of the electron-magnon interaction which is essential in fields such as spintronics and Fe-based superconductivity.

[1] Peter Grünberg Institut, Forschungszentrum Jülich and JARA, 52425 Jülich, Germany. [2] Faculty of Physics and Applied Computer Science, AGH University of Science and Technology, al. Mickiewicza 30, 30-059 Kraków, Poland. [3] Institute for Advanced Simulation, Forschungszentrum Jülich and JARA, 52425 Jülich, Germany. [4] Institute of Scientific and Industrial Research, Osaka University, Ibaraki, Osaka 567-0047, Japan. Correspondence and requests for materials should be addressed to E.M. (email: e.mlynczak@fz-juelich.de)

Spin-flip excitations, including single-particle Stoner and collective spin-wave excitations (magnons), schematically shown in Fig. 1a, are fundamental for the description of ferromagnetic materials[1–4]. The interaction between conduction electrons and magnons is critical for fundamental physical properties, such as the temperature dependence of the resistivity[5] and magnetotransport[6]. It also plays an essential role in models that describe the laser-induced ultrafast demagnetization[7]. On the more applied side, electron-magnon interactions are the basis of the field of magnonics, which offers prospects of faster and more energy-efficient computation[8]. While magnons have a well defined dispersion relation with excitation energies up to a few hundred meV, the Stoner excitations form a quasi-continuum in the magnetic excitation spectrum and their excitation energies are typically in the order of a few eV (Fig. 1b). Although the spin of majority electrons is more likely flipped than that of minority electrons (Fig. 1b), a minority spin flip can have a strong effect on the electronic dispersions, as this article reveals.

Electron dispersion anomalies, such as kinks, are regarded as signatures of an electron-boson interaction, expected to occur at the scale of the boson energy (typically up to few hundred meV)[9–13]. In the case of superconducting materials, the appearance of kinks is a priceless clue pointing to the origin of the electron-electron coupling[14–17]. In ferromagnetic materials, kinks observed by photoemission at binding energies of 100–300 meV were interpreted as originating from the electron-magnon interaction because the involved energy scale was regarded as too large to reflect electron-phonon interaction[12,13,18]. Up to now, this interpretation was merely a suggestion, as no ab initio method has been able so far to reproduce magnon-induced kinks.

In this work, we have experimentally mapped the electronic band structure of an Fe(001) thin film and identified a characteristic kink located 1.5 eV below the Fermi level, which can be reproduced by ab initio calculations based on a diagrammatic expansion of the self-energy, a quantity that describes the deviation of the quasiparticle spectrum from the 'undressed' electron picture. This $GT$ self-energy (Fig. 1c) accounts for the coupling of electrons or holes (Green function $G$) to the correlated many-body system through the creation and absorption of spin excitations ($T$ matrix), taking into account the full nonlocal excitation spectrum with magnons and Stoner excitations treated on an equal footing. The $T$ matrix, which describes the correlated motion of an electron-hole pair with opposite spins, is a mathematically complex quantity because it depends on four points in space (two incoming and two outgoing particles) and time (or frequency). The method is a first-principles approach, therefore, apart from the atomic composition, no additional parameters are used. It naturally takes into account nonlocal electron correlations (momentum dependence of self-energy), which were recently experimentally shown to be important for 3d ferromagnets[19]. Details of the theory are presented in refs. [20,21].

## Results

**Momentum-resolved photoemission.** To get experimental access to the bulk electronic structure of Fe, we have used a thin Fe film (38 ML) deposited on a Au(001) single crystal. The photoemission measurements have been performed at the NanoESCA beamline of Elettra, the Italian synchrotron radiation facility, using a modified FOCUS NanoESCA photoemission electron microscope (PEEM) in the **k**-space mapping mode[22]. The experiment is shown schematically in Fig. 1d. We will discuss here the results obtained using s-polarized light of $h\nu = 70$ eV, which according to the free-electron final state model induces transitions from the initial states located close to the Γ point of the bulk Brillouin zone. Complementary results of the

measurements with p-polarized light, as well as spin-resolved measurements are shown in the Supplementary Figs. 1 and 2 and discussed in the Supplementary Notes 1 and 2.

Figure 2a presents a comparison between experiment and a mean-field band structure of bulk Fe obtained with the local-spin-density approximation (LSDA)[23] of density functional theory (DFT). The blue (red) labels: $\Delta_1$, $\Delta_2$, etc. identify the symmetry of the orbital part of the wavefunctions for minority (majority) states along Fe(001) direction[24]. Here, we neglect the spin-orbit coupling (SOC), as we are interested in the general shape of the bands within a wide binding energy range. The effect of the SOC on the Fe(001) electronic states close to the Fermi level was discussed earlier[25]. In the Supplementary Fig. 1, we also compare the experimental spectrum to $GW$ calculations[26–28], which include quasiparticle renormalization effects beyond DFT. The identification of the experimentally observed electronic states is possible thanks to the results of the spin-polarized measurements (Supplementary Fig. 2) and the consideration of the dipole selection rules that depend on the photon polarization (Supplementary Note 1). Specifically, we identify a minority band of $\Delta_2$ symmetry, which is particularly sharp, especially in contrast to the majority bands (e.g., $\Delta_1$), which become broad and diffuse directly below the Fermi level (Fig. 2a). Importantly, in contrast to the prediction of LSDA and $GW$ calculations, the experimentally observed minority band $\Delta_2$ exhibits a peculiar anomaly near a binding energy of $E_b = 1.5$ eV marked by arrows in Fig. 2a.

**Ab initio calculations.** Figure 2b shows the theoretical spectral function as obtained from the $GT$ calculation summed over the spins on the left and only for the minority spin on the right. We observe a strong renormalization and lifetime broadening of the band structure, in particular, for the majority bands. For example, the majority $\Delta_2$ band loses its quasiparticle character completely below $E_b = 1$ eV, which explains why this band is not visible in the experiment (Fig. 2a) despite favorable dipole selection rules. Such spin dependence of the electron-electron correlation effects is in line with earlier theoretical reports[29–31] and experimental findings[19,32]. In the minority channel, the calculated band dispersions remain relatively sharp. However, the minority $\Delta_2$ band exhibits an anomaly that seems to coincide with the kink observed in the photoemission experiment.

**Experiment vs. theory.** In order to compare the theoretical prediction with the experimental measurement, we have fitted experimental momentum distribution curves (MDC) with a Lorentzian function on a linear background (Fig. 2e) for binding energies between 0.8 and 2.0 eV and superimposed the fitted peak positions on the experimental and theoretical spectral functions in Fig. 2c, d. Both curves, the experimental and the theoretical one, strikingly show the band anomaly at roughly the same energy and momentum. For further analysis, we compare in Fig. 2f the experimental (circles) and theoretical spectral functions (lines) taken at k = 1.3 Å$^{-1}$. We observe a characteristic double-peak structure in both, which indicates a transfer of spectral weight from one branch of the quasiparticle band to another resulting in the appearance of a kink. The Lorentzians fitted to the experimental dispersion can be used to derive the experimental self-energy, which compares remarkably well with the calculated self-energy (see Supplementary Note 3 for the discussion of the self-energy and Supplementary Fig. 3 for the comparison between theoretical and experimental self-energy).

## Discussion

Interestingly, the binding energy at which the anomaly appears is much higher than what one would normally expect for electron-

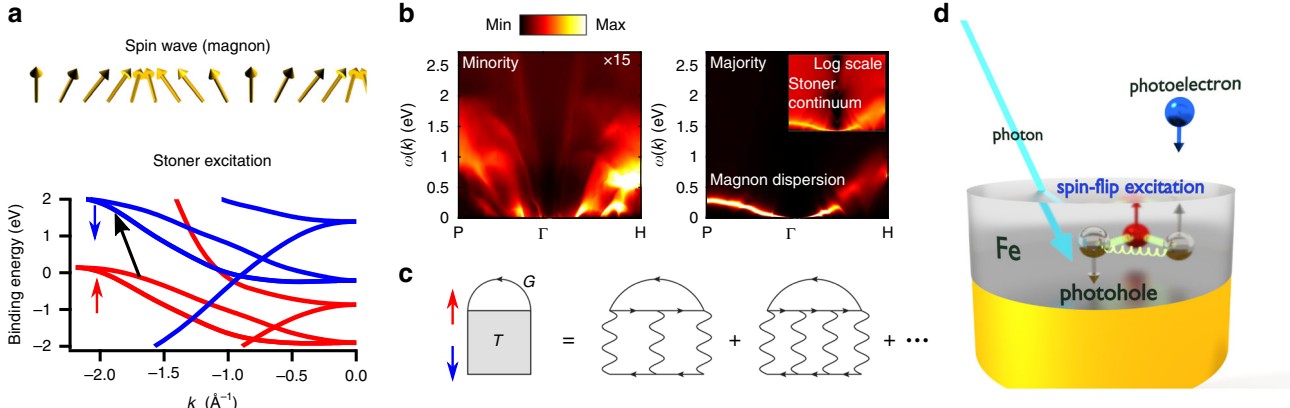

**Fig. 1** Spin-flip effects in bcc Fe and schematics of the theoretical and experimental approach used in the present study. **a** Schematic representation of a collective spin wave and a single-particle Stoner excitation. **b** Calculated momentum and energy dependence of spin excitations in bcc Fe shown separately for minority excitations (left) and majority excitations (right) (spin flip of a minority and majority electron with a change of total spin of +1 and −1, respectively). The color scale on the left image is multiplied by a factor of 15. The inset in the right image presents the same spectrum but plotted in the logarithmic color scale to visualize the Stoner continuum. **c** Diagrammatic expansion of the $GT$ self-energy. Black arrows denote Green functions, and wiggly lines represent the screened interaction. Blue/red arrows denote spin character. **d** Schematics of the spin-resolved momentum microscopy experiment measuring the formation of a bound state consisting of a minority (spin-down) photohole and an electron-hole pair in the majority (spin-up) channel, the photohole and the electron forming a correlated spin-flip excitation

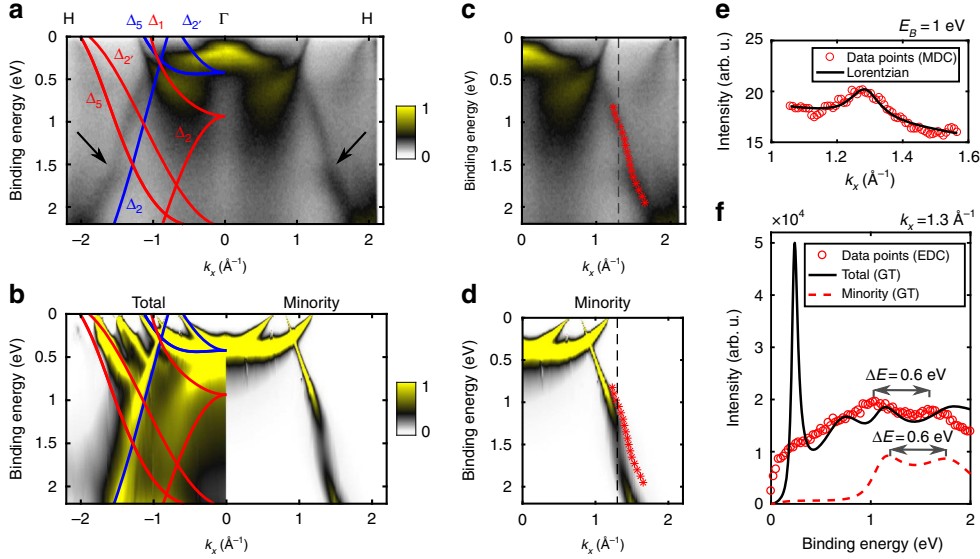

**Fig. 2** Kink far below the Fermi level observed by the photoemission experiment and reproduced by the $GT$ theory. **a** Photoemission measurement performed using $h\nu = 70$ eV and s-polarized light. Superimposed is the result of the single-particle LSDA calculation, where blue and red lines indicate minority and majority bands, respectively. Arrows show the region of the dispersion anomaly. **b** Result of the renormalization with the $GT$ self-energy for both spin directions ($k_x < 0$) together with the undressed LSDA bands and only for the minority spin ($k_x > 0$). Experimental (**c**) and theoretical (**d**) results with superimposed peak positions extracted from the Lorentzian fits (red symbols). **e** Exemplary MDC for $E_b = 1.0$ eV (circles) with a Lorentzian fit (solid line). **f** Total and minority theoretical (black solid and red dashed lines) and total experimental spectral function (red symbols) obtained for $k_x = 1.3$ Å$^{-1}$, marked by dashed vertical lines in **c** and **d**

magnon scattering. Specifically, it is larger than typical magnon energies. The reason for this is twofold. First, self-energy resonances appear not at the boson energy, but rather at the sum of two energies: the boson (e.g., magnon) energy (from $T$) and a single-particle energy (from $G$). Second, we consider a coupling of a propagating minority spin hole to excitations that, due to spin conservation, would have to carry a spin of +1, which is just opposite to the respective spin transfer of magnon excitations. The coupling is thus predominantly with renormalized Stoner

excitations, whose energies are typically larger than magnon energies. In fact, Fig. 1b (left panel) shows a particularly strong resonance around 0.7 eV close to the $H$ point, which, together with a peak in the majority density of states of bulk iron (Supplementary Fig. 4 and Supplementary Note 4) at 0.8 eV, produces a self-energy resonance at around 1.5 eV. This resonance is a manifestation of a broadened many-body state that consists of a majority hole and a superposition of correlated (electron-hole) Stoner excitations and that, by interaction with the minority

band, is ultimately responsible for the appearance of the band anomaly. In other words, a minority photohole is created in the photoemission process, which becomes dressed with electron-hole pairs of opposite spins (Stoner excitations), and flips its spin in the process. This broadened many-body scattering state has a resonance at around $E_b = 1.5$ eV in bcc iron. It bears similarities to the spin polaron[33] in halfmetallic ferromagnets and to the Fermi polaron[34] in ultracold fermion gases.

By examining other **k**-space directions, we find that the appearance of the high-energy kink is very sensitive to the value of the self-energy, and therefore strongly dependent on the **k** direction (see Supplementary Note 5 and Supplementary Fig. 5).

The high-energy kink identified in our work seems similar to the results of F. Mazzola et al.[35,36], who found a kink in the σ band of graphene close to $E_b = 3$ eV. In this case, however, the authors attribute the kink to the strong electron-phonon coupling near the top of the σ band, which effectively places the kink exactly at the boson energy. It is also important to note that some high-energy anomalies, observed especially for cuprates[37–39], were later interpreted to be the result of the photoemission matrix elements[40–42]. We can rule out such an explanation in our experiment, as it is not related to the suppressed photoelectron intensity near a high-symmetry direction[40–42]. It is interesting to note, however, that we observe a similar suppressed intensity for the majority band $\Delta_1$ near the Γ point (Fig. 2a).

While the experimental position and shape of the kink match the prediction by the GT renormalization very well, it should be mentioned that the calculation has been carried out without SOC. The SOC gives rise to an avoided crossing (a spin-orbit gap) between the minority $\Delta_2$ band and both the majority $\Delta_5$ and $\Delta_2'$ bands of the size equal to 60 meV and 100 meV, respectively[25]. However, experimentally, we observe only one kink along the $\Delta_2$ minority band, with the separation in the double-peak structure as large as 600 meV (shown in Fig. 2f), which is why we can rule out that SOC is responsible for the observed band anomaly. Furthermore, the surface states that could potentially interfere with the bulk electronic dispersion are not visible in our experiment (also when measured with other photon energies)[25]. However, to unambiguously prove that the observed kink is not a result of an anticrossing with the surface state, we have analyzed the orbital character of the Fe(001) majority surface state based on relativistic DFT slab calculations (Supplementary Fig. 6). The details of this analysis can be found in the Supplementary Note 6.

The many-body scattering state observed in our experiment can be compared to the 'plasmaron', a bound state of an electron (or hole) and a plasmon[43], which can appear as satellite resonances in photoemission spectra. However, there are important differences, too. First, from a formal point of view, the plasmon propagator is a two-point function (in space and time), while the T matrix is a four-point function obtained from a solution of a Bethe-Salpeter equation. Second, the plasmon energy is quite large (typically around 20 eV), and the plasmaron peaks therefore appear at large binding energies well separated from the quasi-particle bands. In our case, the self-energy resonances are energetically so close to the quasiparticle bands that they strongly interact with each other, potentially leading to anomalous band dispersions like the one discussed in this work.

In summary, our combined experimental and theoretical analysis of the electronic dispersions in iron revealed the formation of a many-body spin flip scattering channel which manifests itself by a kink located at unusually high binding energy ($E_b = 1.5$ eV). This newly discovered excited state of iron is a bound state of a majority hole and a superposition of correlated electron-hole pairs of opposite spins. The observed kink structure is thus of a pure electronic origin, and its prediction from first principles requires a sophisticated quantum-mechanical many-body treatment, in which the k-dependence of the self-energy is sufficiently taken into account.

## Methods

**Momentum-resolved photoemission.** The momentum resolved photoemission was performed using the momentum microscope at the NanoESCA beamline in Elettra synchrotron in Trieste (Italy)[44]. The 38 ML Fe film was grown in-situ on a Au(001) single crystal at low temperature ($T = 140$ K) using molecular beam epitaxy and gently annealed up to 300 °C. This preparation procedure was found previously to result in high-quality Fe(001) films, with no Au present on the Fe surface[25]. This was also confirmed by X-ray photoelectron spectroscopy (XPS) measurements. The microscope is equipped with a W(001)-based spin detector[45], which enables collecting constant energy spin-resolved maps within the entire Brillouin zone of Fe(001). The images were obtained using photon energy of $h\nu = 70$ eV of p or s polarization. The photon beam impinges under an angle of 25 with respect to the sample surface and along the $k_x = 0$ line. According to the free-electron final state model, such a photon energy corresponds to performing a cut through the 3D Brillouin zone close to the Γ point. An analysis of the spin-resolved images was performed following the procedure described in ref. [46]. Before each measurement, the sample was remanently magnetized.

**Ab initio calculations.** The theoretical calculations were performed in the all-electron full-potential linearized augmented-plane-wave (FLAPW) formalism as implemented in the FLEUR DFT and SPEX GW code[27]. To describe the electron-magnon interactions, an ab initio self-energy approximation was derived from iterating the Hedin equations[43], resulting in a diagrammatic expansion from which we have singled out the diagrams that describe a coupling to spin-flip excitations. A resummation of these ladder diagrams to all orders in the interaction yields the GT self-energy approximation, which has a similar mathematical structure as the GW approximation as it is given by the product of the single-particle Green function G and an effective magnon propagator T. The T matrix depends on four points in real space and its implementation involves the solution of a Bethe-Salpeter equation. The numerical implementation is realized using a basis set of maximally localized Wannier functions that allows an efficient truncation of the T matrix in real space. The self-energy is calculated by the method of analytic continuation. The details of the implementation are presented in refs. [20,21].

**Code availability.** The FLEUR code is available at http://www.flapw.de. The SPEX code (http://www.flapw.de/spex) is available from the authors upon request.

## Data availability

All data generated and analyzed during the current study are available from the corresponding author on reasonable request.

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

## Acknowledgements

We thank H. Ibach for valuable discussions. This work was supported by the Helmholtz Association via The Initiative and Networking Fund and by Alexander von Humboldt Foundation.

## Author contributions

E.M., P.G., T.H., M.G., M.J., G.Z., S.S. and V.F. performed experiments with supervision from L.P. and C.T. M.C.T.D.M. and C.F. developed the theoretical method with supervision of S.B. I.A. provided *GW* band structure calculations. G.B. performed the slab calculations. E.M. analyzed experimental data. E.M., M.C.T.D.M. and C.F. wrote the manuscript with contributions from all the co-authors. S.B. and C.M.S. supervised the project.
