## [Peer Review File · Nature Communications]

Reviewers' comments:

Reviewer #1 (Remarks to the Author):

This manuscript addresses the detection and understanding of quasiparticles in magnetic systems. Specifically, the authors address the coupling/interaction involving charge carriers and spin waves in a metallic ferromagnet.

In my opinion, this is a topic of growing interest. Spin waves (magnons) attract interest as entities which can be losslessly transported in materials - hence hinting at the possibility of ultra-low energy logic devices (which utilise spin instead of charge). The interactions involving electrons and magnons are also of fundamental interest; for example by facilitating unconventional forms of superconductivity. Whilst few papers exist in which electron-magnon coupling is understood using ARPES, I'm sure that this will grow in the near future. In short, I think that the present manuscript is novel and timely, and will be well received by the broad readership of Nature Comms.

Regarding the manuscript itself: it is difficult to find anything to be critical about. The manuscript is clear and well written. It is concise, and the conclusions are supported by the high quality data and clear discussions. It is clear that the authors understand the pertinent challenges (i.e. choice of photon energy, matrix elements, spin-selection using light polarisation, etc). The supplementary information is well used: i.e. it contains interesting and important data and calculations, and allows the main manuscript to be kept short and clear.

There are a few very minor points which one can be critical about:

1) when the manuscript is printed on A4 paper, then some of the figures are far too small. The labels and other text are essentially illegible, and the color choices (such as yellow on white in fig 3) are not helpful. I would suggest that the authors try to make the figures larger and clearer wherever possible.

2) there are some minor bugs in the document - especially outside of the main text. For example, in the author contributions, "MCTDM" and "MCTD" are both used, but presumably both refer to the same author. In Ref 20, no source information is given except the year. (note I was unable to trace Ref 20, and the title appears to be closely related to the present work, and published by the same authors. It would have been nice to have been able to investigate the contents of Ref 20 in conjunction with this review!). My guess is that the authors have proof read the language in the main text very carefully, but have been less thorough with the references, contributions, etc. I propose that they just check these sections again before acceptance.

To summarise: This is a very well written manuscript, and the topic is interesting to a broad readership. It is also conducted thoroughly and reliably. I have no hesitation in recommending it for publication in Nature Comms. The only criticisms I have are very minor, and could be quickly addressed by the authors. I look forward to seeing this work published soon.

Reviewer #2 (Remarks to the Author):

In this manuscript, Mlynczak et al. report on the experimental observation of a kink in spin-resolved photo-emission spectra taken in ferromagnetic Fe(001). They interpret their results in terms of a "new" many-body bound state of the photo-hole in the minority spin component with a particle-hole excitation in majority spin Fermi sea. I have several concerns about this manuscript, detailed below. Although I believe that the reported experimental observations warrant publication, I cannot recommend publication of this manuscript in Nature Communications before substantial revisions have been made.

(1) The bound state reported by the authors is not new. In my opinion, what they describe resembles the so-called fermi polaron: There, a mobile impurity particle (from a minority component) interacts with a majority Fermi sea and gets dressed by particle-hole excitations. Such bound states have recently been observed in transition metal dichalcogenides (TMDs), see [Sidler et al., Nature Physics 13, 255–261 (2017)], and were previously studied extensively in ultracold fermi gases, see e.g. [Massignan et al., Rep.Progr.Phys. 77, 034401 (2014)].

Since the authors find the signature for the new many-body state in the minority response, I find the fermi polaron picture more persuasive than the authors interpretation of a bound state between a majority hole and a spin-flip excitation. The authors should comment on this connection to the Fermi polaron problem.

(2) In my view, the discussion of the theoretical methods — which are central for the interpretation of the experimental results as indicators of a many-body bound state — is insufficient. For a general science audience, the paper uses lots of technical jargon from a rather specific subfield of physics without introduction or explanation (Δ_n bands, GW approximation, etc.). The details about the theory which the authors provide in the supplementary, are insufficient to attempt reproducing their results, and confusing since fits to the experimental data are discussed in the same context as the basics of the theory.

The references the authors provide about the theory, [19] and [20], are also insufficient to justify the bare-minimum discussion of the utilized theory: Ref. [19] is a PhD thesis (not published in a peer-review form) and Ref. [20] only consists of the author names and the title of the paper, but no reference to a journal nor to a pre-print server.

For this work to be considered by Nature Communications, the discussion of the theory would have to meet higher standards.

If the authors agree that the fermi polaron perspective is useful, they should also consider providing a comparison of their theoretical calculations to the conceptually much simpler Chevy ansatz wavefunction. This would make the nature of the observed many-body resonance more apparent.

(3) Minor remark:

In the abstract, the authors talk about coupling of electrons to bosons and give ‘polarons’ as an example. What do the authors mean by this? Usually, polarons are electrons dressed by phonons — which are fermionic particles.

Reviewer #3 (Remarks to the Author):

In this interesting manuscript, Dr. Ewa Mlynczak and co-workers, combining ARPES (momentum and spin resolved) data with the first-principles calculations, report the occurrence of the many-body spin flip excitation in the ferromagnetic material (Fe film grown on Au(001) single crystal). The main experiment evidence for such phenomenon is the kink at the minority band, which is positioned at the binding energy (BE) of 1.5 eV. This finding, as the consequence of an electron-magnon interaction, is somehow very fascinating since the kink is positioned at unusual BE which is significantly larger than a typical magnon energy (hundred(s) of meV).

Authors also mentioned the possibility of the spectroscopic signatures appearance related to the surface, which depend on the Fe film preparation conditions (Au diffusions into to the film might also occur). Looking the band dispersion in Fig 2a I have the impression that there is a band bending back at 1.6eV BE. Therefore, an alternative (but less exotic) explanation for the observed kink might reflect a hybridisation between the bulk a surface state bands.

I have few very general remarks/suggestions to the authors.

1. I recommend to the authors perform the curvature method (Review of Scientific Instruments 82, 043712, 2011) to enhance the faint feature at higher BE (1 to 2 eV), located somewhere with k_x between 2 and 0.5 (1/Ångs).

2. I understand that this ARPES studies were performed using only 70 eV photon energy. To rule

out the existence of a surface state the comprehensive AREPS data at different kZ (using different photon energies) must be acquired.

3. The previous published work on the similar system from mostly the same authors (PRX 6, 041048, 2016) reported UPS study combined with XPS. Where was the sample prepared for the synchrotron based experiment? If the sample was grown at the home lab then I strongly suggest authors to describe a sample transfer and how the surface was preserved/refreshed.

4. The authors used the external magnetic field to magnetize the Fe film. Did they accomplish any AREPS measurements without this step? Performing the experiment on the Fe film with many magnetic domains might support the explanation that the observed kink has a magnetic origin.

In summary, the idea of this report is interesting but the manuscript does not present an unambiguous conclusion of the electron-magnon interaction based in the AREPS data. I consider the topic of this manuscript to fit aims and scopes of Nature Communications but to me there is a clear over-interpretation of the data presented here.

I therefore recommend rewriting of the manuscript for submission to Scientific Reports with more extensive discussion including different scenarios for the observed kink together with some of the supplementary material in the main body of the text.

We thank all the reviewers for the critical reading of our manuscript. Below, we discuss all of the issues raised by the reviewers. We improved our manuscript to clarify the points in question. The modified parts of the text are marked by red color. We hope that the current version of the manuscript will be assessed positively.

Since an additional discussion of the possible influence of the surface states on our findings was requested by Reviewer #3, we have performed relativistic slab calculations and included discussion of this aspect in the Supplementary Information. The authors list includes now also Dr. Gustav Bihlmayer, who performed the slab calculations.

We have modified the format of the article by including not-referenced abstract and modifying the format of the introduction to comply with the Nature Communications format. This results in a new order of the first 18 references.

We have added new references: [19], [24], [26], [28], [31-34] in the main article and [6] and [7] in the Supplementary Information.

Below we discuss in detail all the remarks of the Reviewers:

Remarks of Reviewer #1 This manuscript addresses the detection and understanding of quasiparticles in magnetic systems. Specifically, the authors address the coupling/interaction involving charge carriers and spin waves in a metallic ferromagnet. In my opinion, this is a topic of growing interest. Spin waves (magnons) attract interest as entities which can be losslessly transported in materials - hence hinting at the possibility of ultra-low energy logic devices (which utilise spin instead of charge). The interactions involving electrons and magnons are also of fundamental interest; for example by facilitating unconventional forms of superconductivity. Whilst few papers exist in which electron-magnon coupling is understood using ARPES, I'm sure that this will grow in the near future. In short, I think that the present manuscript is novel and timely, and will be well received by the broad readership of Nature Comms. Regarding the manuscript itself: it is difficult to find anything to be critical about. The manuscript is clear and well written. It is concise, and the conclusions are supported by the high quality data and clear discussions. It is clear that the authors understand the pertinent challenges (i.e. choice of photon energy, matrix elements, spin-selection using light polarisation, etc). The supplementary information is well used: i.e. it contains interesting and important data and calculations, and allows the main manuscript to be kept short and clear. There are a few very minor points which one can be critical about: 1) when the manuscript is printed on A4 paper, then some of the figures are far too small. The labels and other text are essentially illegible, and the color choices (such as yellow on white in fig 3) are not helpful. I would suggest that the authors try to make the figures larger and clearer wherever possible.

Authors: To address this issue we have combined Figure 2 and Figure 3 into a single 2-column wide Figure. We have included minor changes to the new Figure 2: i) we changed the aspect ratio of the figures to make it consistent for all the figures ii) we improved graphics/colors to make the figures better legible also when printed iii) the exemplary Lorentzian line is now shown before subtraction of the linear background (new Fig. 2 e).

2) there are some minor bugs in the document - especially outside of the main text. For example, in the author contributions, "MCTDM" and "MCTD" are both used, but presumably both refer to the same author. In Ref 20, no source information is given except the year. (note I was unable to trace Ref 20, and the title appears to be closely related to the present work, and published by the same authors. It would have been nice to have been able to investigate the contents of Ref 20 in conjunction with this review!). My guess is that the authors have proof read the language in the main text very carefully, but have been less thorough with the references, contributions, etc. I propose that they just check these sections again before acceptance.

Authors: We have corrected a mistake in the author contributions section. Reference 20 is a manuscript which describes the details of the theoretical method we used, currently in review in Phys. Rev. B. and available on the arXiv preprint server: [arXiv:1809.02395](https://arxiv.org/abs/1809.02395). We have also corrected small typos in the bibliography of the supplement.

To summarise: This is a very well written manuscript, and the topic is interesting to a broad readership. It is also conducted thoroughly and reliably. I have no hesitation in recommending it for publication in Nature Comms. The only criticisms I have are very minor, and could be quickly addressed by the authors. I look forward to seeing this work published soon.

Remarks of Reviewer #2: In this manuscript, Mlynczak et al. report on the experimental observation of a kink in spin-resolved photo-emission spectra taken in ferromagnetic Fe(001). They interpret their results in terms of a "new" many-body bound state of the photo-hole in the minority spin component with a particle-hole excitation in majority spin Fermi sea. I have several concerns about this manuscript, detailed below. Although I believe that the reported experimental observations warrant publication, I cannot recommend publication of this manuscript in Nature Communications before substantial revisions have been made.

Authors: We reply to the more general comment (2) first and then to comment (1).

(2) In my view, the discussion of the theoretical methods — which are central for the interpretation of the experimental results as indicators of a many-body bound state — is insufficient.

For a general science audience, the paper uses lots of technical jargon from a rather specific subfield of physics without introduction or explanation (Δ_n bands, GW approximation, etc.).

The details about the theory which the authors provide in the supplementary, are insufficient to attempt reproducing their results, and confusing since fits to the experimental data are discussed in the same context as the basics of the theory.

The references the authors provide about the theory, [19] and [20], are also insufficient to justify the bare-minimum discussion of the utilized theory: Ref. [19] is a PhD thesis (not published in a peer-review form) and Ref. [20] only consists of the author names and the title of the paper, but no reference to a journal nor to a pre-print server. For this work to be considered by Nature Communications, the discussion of the theory would have to meet higher standards.

Authors: We thank the Referee for pointing out lacking information about the theory. We have now included some additional, descriptive details of the theory to make the present paper more self-contained. For further details, we would like to refer the reader to an accompanying theory paper, which has been submitted to PRB and which we have made available on arxiv:1809.02395.

The theoretical approach relies on many-body perturbation theory and is a first-principles approach. This means that, apart from the atomic composition (bcc iron), there is no additional parameter. The electrons move in the electrostatic potential of the atomic nuclei (positive point charges), so they do not form a homogeneous, free fermionic gas. Close to the atomic nuclei, the electrons can acquire "relativistic" velocities. Here, we use the Dirac equation. The single-particle wave functions then consist of four-component spinors consisting of a large and a small component in each spin channel. The single-particle wave functions are calculated within density-functional theory (DFT), which gives a realistic description for the electronic ground state. For the calculation of the photoelectron spectra (excited states!), however, we have to go beyond DFT and use a diagrammatic technique within many-body perturbation theory. The diagrams are derived from Hedin's equations, which allow a systematic expansion of the electronic self-energy. The first-order self-energy approximation derived in this way can be written as the product of the Green function (G) and the screened interaction (W), giving the so-called GW approximation, to which the Referee has referred as "technical jargon". We would respectfully disagree with using the word "jargon" here. (The GW method is nowadays a routine method used in many theoretical condensed-matter studies. There is also a wikipedia page about the GW approximation.) However, we have added a short description and references about GW in the resubmitted manuscript to address a broader audience.

In order to describe higher-order scattering events, in particular, the scattering of electrons and magnons, we have to go beyond GW by including so-called vertex corrections, which leads us to what we have termed "GT self-energy". (Its definition is again motivated by Hedin's self-energy expansion including an infinite resummation of ladder diagrams.) We want to emphasize that the GT approach is a first-principles approach. No extra parameters except for the atomic composition are used in the calculation. The system is infinite with periodic boundary conditions. (By using interpolation techniques in reciprocal space, the boundaries can be pushed to infinity, thereby treating an infinite number of electrons.) All electrons (spin up and spin down) interact with each other via the Coulomb interaction ($\sim 1/r$) and are treated quantum mechanically. Exchange and correlation effects, electronic screening and lifetime broadening are taken into account.

In this context, it is interesting to note that we had performed the theoretical calculation first. The appearance of the kink in the theoretical spectrum has motivated the experimentalists to investigate

this band in detail. In fact, they then found a kink in the same band at roughly the same energy and momentum. So, it was a genuine theoretical prediction.

We have added the following additional explanation to the main part of the manuscript:

- "The T matrix, which describes the correlated motion of an electron-hole pair with opposite spins, is a mathematically complex quantity because it depends on four points in space (two incoming and two outgoing particles) and time (or frequency). The method is a first-principles approach, therefore apart from the atomic composition, no additional parameters are used. It naturally takes into account nonlocal electron correlations (momentum dependence of self-energy), which were recently experimentally shown to be important for 3d ferromagnets [19]."

And included the following modifications (additions are in red):

- "the local-density approximation (LSDA) [21] of density functional theory (DFT)" - "[...] we also compare the experimental spectrum to GW calculations [26-28], which include quasiparticle renormalization effects beyond DFT."

- "Superimposed is the result of the single-particle LSDA calculation" (in Caption of Fig.2)

The Referee has also criticized the usage of the expression " Δ_n bands" without explanation. These labels are used in electronic-structure theory to distinguish different electronic bands and label symmetry groups (irreducible representations of group theory). We think that these labels are widely known in the community, however to address a broad audience we have added an additional reference [23] and included small modifications to the sentence that explains the labels: "The blue (red) labels (Δ_1 , Δ_2 , etc.) identify the symmetry of the orbital part of the wavefunctions for minority (majority) states along Fe(001) direction [23]."

In the Supplemental Information, we discuss the comparison of the self-energy derived from the experiment and resulting from our theoretical approach. Since both compare well, we find this comparison an additional proof of the correctness of our theoretical approach and a virtue of the manuscript rather than a weak point. However, to clearly separate the description of the theoretical and experimental self-energy we have added the following sentences:

-> "The theoretical spin- σ spectral function of the interacting many body system reads (...)"

-> "The theoretical self-energy can be compared to the experimental self-energy, whose imaginary part is defined as: (...)"

In the 'Self-energy' section of the Supplemental Information we have also added an additional explanation of the GT self-energy diagrams:

"The GT self-energy diagrams are shown in Fig. 1d in the main part of the article. From iterating the Hedin equations it follows that the lowest-order diagram is of third order in the screened interaction W , for which we use the random-phase approximation. The Bethe-Salpeter equation for the T matrix is formulated in such a way that it yields the ladder diagrams to all orders starting from the third-order

diagram. Summing to all orders is necessary to obtain well-defined collective spin excitations and realistic renormalization effects in the Stoner continuum. The self-energy is calculated on an imaginary-frequency mesh and then analytically continued to the full complex plane using the Padé approximation.”

(1) The bound state reported by the authors is not new. In my opinion, what they describe resembles the so-called fermi polaron: There, a mobile impurity particle (from a minority component) interacts with a majority Fermi sea and gets dressed by particle-hole excitations. Such bound states have recently been observed in transition metal dichalcogenides (TMDs), see [Sidler et al., Nature Physics 13, 255–261 (2017)], and were previously studied extensively in ultracold fermi gases, see e.g. [Massignan et al., Rep.Progr.Phys. 77, 034401 (2014)]. Since the authors find the signature for the new many-body state in the minority response, I find the fermi polaron picture more persuasive than the authors interpretation of a bound state between a majority hole and a spin-flip excitation. The authors should comment on this connection to the Fermi polaron problem. If the authors agree that the fermi polaron perspective is useful, they should also consider providing a comparison of their theoretical calculations to the conceptually much simpler Chevy ansatz wavefunction. This would make the nature of the observed many-body resonance more apparent.

Authors: There is also the term "spin polaron", which was discussed for halfmetallic ferromagnets in [Irkhin et al, J. Phys. Condens. Matt. 19, 315201 (2007)]. There, the interaction is with collective (spin-wave) excitations. In the Fermi polaron picture, the interaction is with single-particle (Stoner) excitations. Our calculation treats both collective and renormalized Stoner excitations on the same footing, since the electrons (also the majority electrons) interact with each other. (This leads, for example, to a profound shift of quasiparticle weight in the Stoner continuum and also to lifetime effects of the collective spin waves.) So, our method takes both effects into account, and we think that using terms like "spin polaron" or "Fermi polaron" would be a misleading simplification. Therefore, we prefer to describe the state without giving it an explicit name. However, we have added the following sentence to the manuscript:

“[This broadened many-body scattering state] bears similarities to the spin polaron [Irkhin et. al.] in halfmetallic ferromagnets and to the Fermi polaron [Massignan et. al.] in ultracold fermion gases.”

Since the Referee asked us explicitly about the Chevy ansatz, we would like to comment on that. The Chevy ansatz gives the many-body wave function of the excited state as a linear combination of an unrenormalized (minority) impurity state (added to the electronic ground state) with three-particle states consisting of (majority) holes and unrenormalized Stoner excitations. We use a Green-function technique instead of approximating the many-body wave function. Therefore it is difficult to directly compare the two approaches. However, we can deduce from the diagrams through which excited states the propagating particle (described by the Green function) evolves. In our case, we perform a "one-shot" renormalization with respect to the GT self-energy, which would correspond to the Chevy ansatz with the difference that the electron-hole pairs (linear term of Chevy wave function) perform a correlated motion (because the fermions interact with each other) and can, for example, form collective excitations. Furthermore, again due to the interaction in the majority channel, we can have spontaneous

spin fluctuations in the ground state (analogous to vacuum fluctuations in QED), with which the propagating particle can interact. In this case, we treat higher-order terms in the many-body wave function not included in the Chevy ansatz. These additional features are important because they lead to collective excitations and important renormalization effects in the Stoner continuum, without which a quantitative agreement with experiment cannot be obtained. In this context, it is also important to note that the Fermi polarons are unique eigensolutions of the many-body system, while we deal with a superposition of infinitely many eigenstates in a relatively broad energy range. As a result, we observe a kink with profound lifetime effects rather than an avoided crossing of two polaron branches (attractive and repulsive). All that being said, we would like to point out that we do not claim the scattering state to be "new" in the sense that similar many-body states have never been discussed before (for example, based on model calculations). In the title, we write "new electron-magnon scattering channel in Fe". Here, the emphasis is on "scattering channel in Fe". We do not see it as a new scattering phenomenon. But never have such states been explicitly shown (e.g. by photoemission) for a ferromagnetic material and, at the same time, described theoretically using an ab initio method in quantitative agreement with experiment. We have revised the text accordingly:

- "superposition of **renormalized** spin-flip excitations".- We have removed the word "exotic" from "exotic many-body state"

- "[...] manifestation of a many-body state that can be described as a majority hole bound to correlated (electron-hole) Stoner excitations"-> "[...] manifestation of a **broadened** many-body state that **consists of a majority hole and a superposition of** correlated (electron-hole) Stoner excitations"

- "This **broadened** many-body **scattering** state has a resonance at around $E_b = 1.5$ eV **in bcc iron**" (the word "new" was removed)

- "This newly discovered excited state **of iron** is a [...]"

(3) Minor remark: In the abstract, the authors talk about coupling of electrons to bosons and give 'polarons' as an example. What do the authors mean by this? Usually, polarons are electrons dressed by phonons — which are fermionic particles.

Authors: It is true that the word "polarons" does not fit here. It has been replaced by the word "plasmons" in the new manuscript.

Remarks of Reviewer #3: In this interesting manuscript, Dr. Ewa Mlynczak and co-workers, combining ARPES (momentum and spin resolved) data with the first-principles calculations, report the occurrence of the many-body spin flip excitation in the ferromagnetic material (Fe film grown on Au(001) single crystal). The main experiment evidence for such phenomenon is the kink at the minority band, which is positioned at the binding energy (BE) of 1.5 eV. This finding, as the consequence of an electron-magnon interaction, is somehow very fascinating since the kink is positioned at unusual BE which is significantly larger than a typical magnon energy (hundred(s) of meV). Authors also mentioned the possibility of the spectroscopic signatures appearance related to the surface, which depend on the Fe film preparation conditions (Au diffusions into to the film might also occur). Looking the band dispersion in

Fig 2a I have the impression that there is a band bending back at 1.6eV BE. Therefore, an alternative (but less exotic) explanation for the observed kink might reflect a hybridisation between the bulk and surface state bands.

Authors: To address this issue we have performed additional slab calculations to analyze position (in k and binding energy) of possible surface states as well their spin and orbital character. With these data at hand we are able to unambiguously prove that even if the surface state at the discussed binding energy and momentum existed on the surface of our sample it would simply cross the minority band of Δ_2 orbital symmetry (no anticrossing is expected). We also note on the experimental side, that we have not observed surface states on Fe(001) surface, which we already discussed in the previous paper (PRX 6, 041048, 2016). We have added an additional text in the main article: **However, to unambiguously prove that the observed kink is not a result of an anticrossing with the surface state, we have analyzed the orbital character of the Fe(001) majority surface state based on relativistic DFT slab calculations. The details of this analysis can be found in the Supplementary Information (Fig. S6).**

I have few very general remarks/suggestions to the authors. 1. I recommend to the authors perform the curvature method (Review of Scientific Instruments 82, 043712, 2011) to enhance the faint feature at higher BE (1 to 2 eV), located somewhere with k_x between 2 and 0.5 (1/Å).

Authors: We know about the curvature method and we also tried to apply it for this set of data which however did not bring any new insights into the analysis. The commonly used method of analysis of the band dispersions (also used in our manuscript) is fitting of the MDC with Lorentzian profiles which allows quantitative analysis in terms of the self-energy.

2. I understand that this ARPES studies were performed using only 70 eV photon energy. To rule out the existence of a surface state the comprehensive ARPES data at different k_z (using different photon energies) must be acquired.

Authors: As mentioned above, in the extensive studies (also using different photon energies) of Fe(001) surface using ARPES we have not observed signatures of the surface states.

3. The previous published work on the similar system from mostly the same authors (PRX 6, 041048, 2016) reported UPS study combined with XPS. Where was the sample prepared for the synchrotron based experiment? If the sample was grown at the home lab then I strongly suggest authors to describe a sample transfer and how the surface was preserved/refreshed.

Authors: The samples were always freshly in-situ grown films. Immediately after growth the sample was transferred from the preparation chamber to the photoemission microscope chamber without breaking the vacuum.

4. The authors used the external magnetic field to magnetize the Fe film. Did they accomplish any ARPES measurements without this step? Performing the experiment on the Fe film with many magnetic domains might support the explanation that the observed kink has a magnetic origin.

Authors: The experiments were always performed after magnetizing the sample. Even if we would collect photoelectrons from more than one magnetic domain, the kink related to the formation of the many body state discussed in this work would still be observed.

In summary, the idea of this report is interesting but the manuscript does not present an unambiguous conclusion of the electron-magnon interaction based in the AREPS data. I consider the topic of this manuscript to fit aims and scopes of Nature Communications but to me there is a clear over-interpretation of the data presented here.

Authors: We cannot agree with the statement on 'over-interpretation' of data. We performed a rigorous analysis of the experimental spectra by fitting the momentum distribution curves out of which we were able to derive the parameters that are related to the self-energy of the system. In this analysis, we followed a procedure that is widely used in the literature. Such obtained values agree exceptionally well with the results that we independently obtained based on the *ab initio* calculations. The interpretation that we present in the manuscript is based on the careful analysis of all the theoretical and experimental aspects that could play a role in the discussed effects.

I therefore recommend rewriting of the manuscript for submission to Scientific Reports with more extensive discussion including different scenarios for the observed kink together with some of the supplementary material in the main body of the text.

Authors: We hope that the improvements that we made in the text of the article, additional supplemental material and discussion presented in this response convince the reviewer that our work is suitable for publication in Nature Communications.

REVIEWERS' COMMENTS:

Reviewer #2 (Remarks to the Author):

I have read the revised version of the manuscript, in which the authors report the observation of a kink in spin-resolved ARPRES spectra which they interpret as a result of the formation of a complex many-body bound state of a photo-hole with collective spin-wave and Stoner excitations, both involving spin flips. The interpretation of the data relies on a remarkable agreement which the authors obtain with theoretical ab-initio calculations.

The authors have addressed the concerns voiced in my first referee report, and their referee reply has clarified a lot of my questions. Most importantly, the authors have made their accompanying theoretical work available on the arXiv, which describes in detail the advanced diagrammatic methods they are using. I still find the manuscript rather hard to follow, since, for my taste, it uses too many theoretical terms which are difficult to understand without some deeper knowledge of the field or the underlying diagrammatic calculations. However, I find the reported agreement of the measurement with the authors theoretical calculations very impressive. After reading the authors reply to my referee report, I also find the authors interpretation of their data convincing. Given the relevance of this work in the context of understanding electron-magnon interactions in more detail, I would now recommend publication of this work in Nature Communications.

Optionally, I would suggest to the authors to include some of their very clear explanations in their reply to my referee report into their paper. For example, I found the discussion of the Chevy ansatz, and the explanation how the GT method goes beyond this approximation, very useful. Similarly, I found the discussion of how the many-body bound state found by the authors differs from both a Fermi polaron and a spin polaron extremely useful. While the authors added a comment in the text, referencing Fermi and spin polarons briefly, I certainly think that adding a brief discussion, as in their reply, would significantly improve the manuscript.

Reviewer #2 (Remarks to the Author):

I have read the revised version of the manuscript, in which the authors report the observation of a kink in spin-resolved ARPRES spectra which they interpret as a result of the formation of a complex many-body bound state of a photo-hole with collective spin-wave and Stoner excitations, both involving spin flips. The interpretation of the data relies on a remarkable agreement which the authors obtain with theoretical ab-initio calculations.

The authors have addressed the concerns voiced in my first referee report, and their referee reply has clarified a lot of my questions. Most importantly, the authors have made their accompanying theoretical work available on the arXiv, which describes in detail the advanced diagrammatic methods they are using. I still find the manuscript rather hard to follow, since, for my taste, it uses too many theoretical terms which are difficult to understand without some deeper knowledge of the field or the underlying diagrammatic calculations. However, I find the reported agreement of the measurement with the authors theoretical calculations very impressive. After reading the authors reply to my referee report, I also find the authors interpretation of their data convincing. Given the relevance of this work in the context of understanding electron-magnon interactions in more detail, I would now recommend publication of this work in Nature Communications.

Optionally, I would suggest to the authors to include some of their very clear explanations in their reply to my referee report into their paper. For example, I found the discussion of the Chevy ansatz, and the explanation how the GT method goes beyond this approximation, very useful. Similarly, I found the discussion of how the many-body bound state found by the authors differs from both a Fermi polaron and a spin polaron extremely useful. While the authors added a comment in the text, referencing Fermi and spin polarons briefly, I certainly think that adding a brief discussion, as in their reply, would significantly improve the manuscript.

We thank the Reviewer for the positive assessment of our manuscript. The reviewer mentioned that as an option we could include more of the details describing the theoretical method that we discussed in the answer to the first review. We decided to keep the original, concise form of the manuscript and in the same time to opt for publishing the review and our response together with the article. In this way, an interested reader can find more of the details in these materials.